# Exploring Influence of Sampling Strategies on Event-Based Landslide Susceptibility Modeling

**Jhe-Syuan Lai [1,2], Shou-Hao Chiang [2,3]**  **and Fuan Tsai [2,3,*]**

1   Department of Civil Engineering, Feng Chia University, Taichung 40724, Taiwan
2   Department of Civil Engineering, National Central University, Taoyuan 32001, Taiwan
3   Center for Space and Remote Sensing Research, National Central University, Taoyuan 32001, Taiwan
*   Correspondence: ftsai@csrsr.ncu.edu.tw; Tel.: +886-3-4227151 (ext. 57619)

**Abstract:** This study explores two modeling issues that may cause uncertainty in landslide susceptibility assessments when different sampling strategies are employed. The first issue is that extracted attributes within a landslide inventory polygon can vary if the sample is obtained from different locations with diverse topographic conditions. The second issue is the mixing problem of landslide inventory that the detection of landslide areas from remotely-sensed data generally includes source and run-out features unless the run-out portion can be removed manually with auxiliary data. To this end, different statistical sampling strategies and the run-out influence on random forests (RF)-based landslide susceptibility modeling are explored for Typhoon Morakot in 2009 in southern Taiwan. To address the construction of models with an extremely high false alarm error or missing error, this study integrated cost-sensitive analysis with RF to adjust the decision boundary to achieve improvements. Experimental results indicate that, compared with a logistic regression model, RF with the hybrid sample strategy generally performs better, achieving over 80% and 0.7 for the overall accuracy and kappa coefficient, respectively, and higher accuracies can be obtained when the run-out is treated as an independent class or combined with a non-landslide class. Cost-sensitive analysis significantly improved the prediction accuracy from 5% to 10%. Therefore, run-out should be separated from the landslide source and labeled as an individual class when preparing a landslide inventory.

**Keywords:** cost-sensitive analysis; landslide susceptibility; random forests; sampling strategy; Typhoon Morakot

## 1. Introduction

In 2009, Typhoon Morakot brought extensive rainfall, causing numerous landslides in southern Taiwan. An official report [1] documented that 769 individuals died or went missing directly or indirectly because of these landslides, and that approximately US$526 million was lost due to damage to the agriculture, forestry, and fishery industries. Of particular note, a riverside village called Xiaolin (sometimes spelled Shiaolin or Hsiaolin) was destroyed by the landslides from a devastating landslide nearby, which led to approximately 500 fatalities. After this event, a large number of studies have concentrated on detecting, characterizing, assessing, and modeling landslide events in areas such as Xiaolin Village and the Kaoping watershed in the interest of identifying methods to mitigate or minimize the effect of similar disasters in the future. For example, Mondini et al. [2] and Mondini and Chang [3] have utilized spectral and geo-environmental information to detect landslide areas. Deng et al. [4] and Tsai et al. [5] have determined the extent and analyzed the topographic and environmental characteristics of landslides using satellite images and spatial analysis. From a geological or geotechnical viewpoint, Tsou et al. [6] and Wu et al. [7] have explored the relationship

between rainfall duration, geological structures, soil types, geo-morphological features, and other similar variables. In addition, Chen et al. [8] estimated the average landslide erosion rate, and Chang et al. [9] modeled the spatial occurrence of landslides between 2001 and 2009.

Modeling landslide susceptibility is a critical and forward task within the framework of landslide hazard and risk management [10–12]. It is used to assess the likelihood (0 to 1) or degree (e.g., low, moderate, and high) of landslide occurrence in an area with given local terrain attributes [13]. Traditionally, modeling methods can be classified into three main categories [14–16] of approaches: deterministic [17–19], heuristic [20,21], and statistical [22–27]. A review of the literature indicates that continuing improvements in remote sensing and geographic information systems (GIS) have led to the incorporation of machine learning (and data mining) models for the evaluation of regional landslide susceptibility; examples include decision tree [28–30], rough set [31,32], support vector machine [16,33], neural network [16,34–43], fuzzy theory [35,44–48], neural fuzzy systems [35,42,49–51], and entropy- and evolution-based algorithms [15,38,52,53]. Some related works have also considered using various composite strategies based on previous approaches to achieve a specific purpose [45,47,54]. A number of studies in the literature have compared different methods and various results obtained from modeling landslide susceptibility for different study sites and collected data [16,30,35,38,42–44,55–57]. However, no general consensus has yet been reached concerning which is the best procedure and algorithm for evaluating landslide susceptibility [57,58].

With improved temporal, spatial, and spectral resolutions of remote sensing observations, the automatic or semiautomatic approaches are widely used to detect landslide areas during a single triggering event using pixel-based [2,3,59–61] and object-oriented [62–65] strategies to produce landslide inventories (databases). Then, the susceptibility assessment using an event-based landslide inventory [25,66], landslide hazard, and vulnerability and risk assessments in the framework of landslide hazard and risk management can be further made [66]. To maintain the quality of these subsequent tasks, landslide susceptibility modeling must be studied and explored further.

Petschko et al. [67] identified three uncertainty issues when constructing data-driven landslide susceptibility models (i.e., statistical and machine-learning-based approaches): input data, constructed models, and susceptibility maps. Landslide inventory and factors are the major input data. The spatial resolution, scale, type, and accuracy of the landslide inventory, and various landslide related factors are also important [12]. Some input data may be produced subjectively, and the experience of the digitizer and measurement errors or imprecision in data processing can be sources of parametric uncertainty [11,12,68–70]. Therefore, the quality assurance/quality control (QA/QC) assessment of all input data is an essential part of the process of reducing data uncertainties. Brenning [71] and Guzzetti et al. [72] have discussed some cases for the evaluation of the uncertainty of landslide susceptibility models. Statistical indices commonly used to verify a model or classifier performance include overall accuracy, kappa coefficient, precision (also called user's accuracy (UA)), recall (also called producer's accuracy (PA)), and receiver operating characteristics (ROCs) curve. The quality of a landslide susceptibility map depends on the input data and selected algorithm or model. The mapping results are also affected by the analytical unit (e.g., grid cell, slope, or terrain units) and thresholds and number of degrees for discretizing the likelihood of landslide occurrences [67,73].

From a geotechnical viewpoint, three major features are typical of natural terrain landslides, and these were highlighted by Dai and Lee [74]. The source area is defined as the surface of the rupture comprising the main scarp and the scarp floor. The landslide trail downslope of the source area is predominately produced by transport of the landslide mass, although erosion and deposition may also occur. The deposition fan is where the majority of the landslide mass is deposited. The term run-out generally describes the downslope displacement of failed geo-materials by landslides and is used to indicate the landslide trail and deposition fan [2]. In general, landslide areas detected by automatic and semiautomatic algorithms from remotely-sensed images might include run-out regions, unless these have been removed manually by comparison with stereo aerial photos and other auxiliary data. However, these run-out areas should be excluded from the landslide area because they are caused by a

different mechanism. Mixing landslide areas with run-out features might reduce the reliability of a landslide susceptibility model.

Selecting a suitable sampling strategy is crucial for transforming the landslide inventory and factors into formats that can be imported as samples to construct a data-driven landslide susceptibility model. It also affects the subsequent results and maps. Hussin et al. [75] summarized the four sampling strategies most commonly used to extract landslide samples for data-driven landslide susceptibility modeling and mapping. These include, (1) the sampling of single pixels extracted from each landslide inventory polygon using the centroid or statistical method [76,77]; (2) extraction of all the pixels within landslide inventory polygons [67], which increases computational loading in modeling; (3) selection of pixels on and around the landslide crown-line, which is called the main scrap upper edge (MSUE) approach [14]; and (4) choosing the pixels within a buffer polygon around the upper landslide scarp area, which is referred to as the seed-cell approach [77] and was proposed by Suzen and Doyuran [78]. Several studies have compared the effectiveness of various sampling strategies [77,79,80] and concluded that there could be some extraction and registration problems when using centroid pixels [75]. In addition, results based on the MSUE and seed-cell approaches might be subjective and performed manually in some cases.

Selecting a proper algorithm to construct a landslide susceptibility model is also a critical task. The random forests (RF) algorithm has received increasing attention due to (1) excellent accuracy [81], (2) fast processing [82,83], (3) few parameter settings [84,85], (4) high-dimension data analysis ability [86,87], and (5) insensitivity to imbalanced training data [88]. Fewer studies have investigated the performance of the RF algorithm in landslide susceptibility modeling [71,89] than have evaluated the performance of statistical methods such as logistic regression. On the other hand, to address the problem of unbalanced results, the construction of models with extreme false alarm (commission error) or missing (omission error) predictions, the cost-sensitive analysis can be applied to adjust the decision boundary and improve modeling [90–92] (i.e., the cut value that determines landslide and non-landslide).

This study explored two modeling issues that may cause uncertainty in landslide susceptibility assessment when different sampling strategies are applied. The first issue is that extracted attributes within a landslide polygon can be varied by taking the sample from different topographic locations. Ideally, a landslide sampling point represents the geo-environmental characteristics at the location of the landslide initiation point. Previous studies have used different sampling strategies, such as taking samples from the centroid point of landslide polygons, but the effect has not been clarified yet. The second issue is the mixing problem of landslide inventory; here, landslide features detected by automatic or semiautomatic algorithms from remotely-sensed data generally include source and run-out features, unless the run-out portion can be removed manually with stereo aerial photo interpretation and other auxiliary data. The hill slope mechanism of run-out involves transportation and deposition, which differ from slope failure and erosion from landslides. For modeling purposes; therefore, a landslide model constructed using mixed samples may cause uncertainty. The landslide event induced by Typhoon Morakot in 2009 in southern Taiwan was selected as the study case. The definition of landslide in this study describes the downslope movement of soil, rock, and organic masses under the influence of gravity, as well as the landform that results from such movement [93], ignoring the size of the materials.

Two experiments were designed to explore these sampling issues. The first experiment entailed the application of different sampling operators for the extraction of attributes from the centroid, maximum slope, median slope, and minimum slope of landslide and run-out. In the second experiment, this study categorized the run-out samples into the following: (1) An independent (run-out) class, (2) a run-out combined with landslide (source) class, and (3) a run-out combined with non-landslide class. RF was selected to perform landslide susceptibility modeling with geo-spatial data (i.e., remote sensing and GIS data) on the basis of the grid mapping unit, which is a popular and simple analytic format. In addition, this study combined cost-sensitive analysis with RF to adjust the decision boundary of constructed models and thereby reduce extremely high false alarm or missing errors.

## 2. Materials and Methods

### 2.1. Study Site and Data

Covering an area of 117 km$^2$, the Laonong River watershed, located in the Kaoping River watershed in southern Taiwan, is selected as the study site (Figure 1). The elevation of the study site ranges from 258 to 1666 m above sea level, measured from the digital elevation model (Figure 2a), as edited by Chiang et al. [94]. The average slope and standard deviation are 25.84° and 11.98°, respectively. According to the geological and soil maps published by the Central Geological Survey of Taiwan, there are three geological formations and four soil types in this study area. The geological formations are the Lushan, Snhsia, and Toukoshan formations (Figure 2b), and the main soil types (agriculture- and geology-based classification) are alluvium, colluviums, lithosol, and loam soils (Figure 2c).

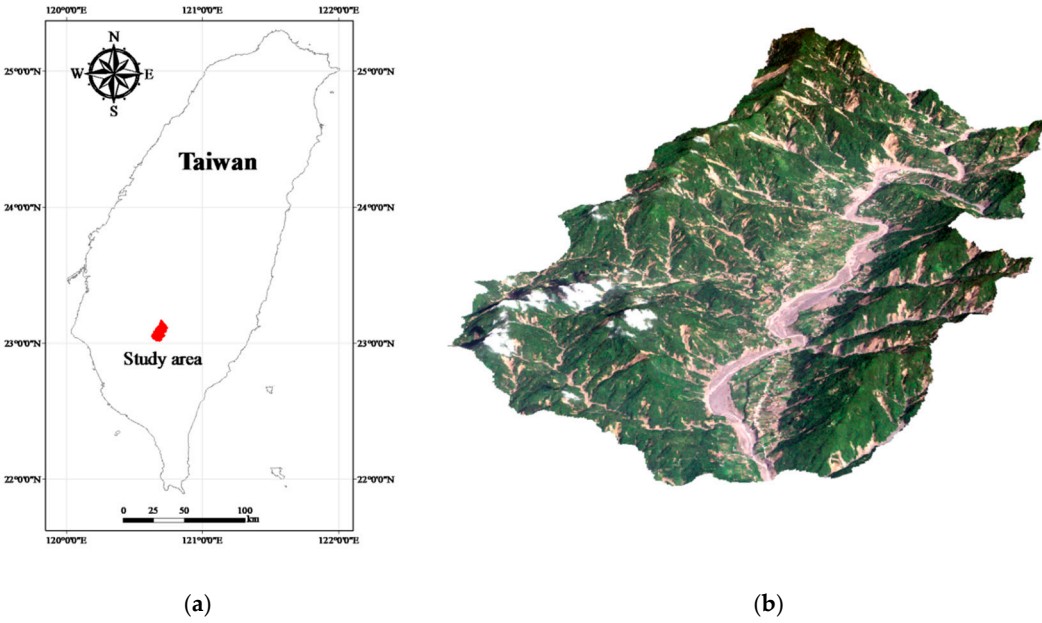

(**a**)　　　　　　　　　　　　　　　　　　　　　　(**b**)

**Figure 1.** Study area: (**a**) Location, (**b**) 3D view from the south.

The landslide inventory map of deep-seated landslides as shown in Figure 3 was interpreted manually after Typhoon Morakot to identify the source and run-out and channel classes from stereo aerial photos and auxiliary data. Detailed information for the landslide inventory is presented in Table 1. This study also collected data on faults, rivers, and roads (Figure 2d), as well as normalized difference vegetation index (NDVI) information (Figure 2e) derived from a pre-event Formosat-2 satellite image. Moreover, all vector-based line features (i.e., distance to fault, river, and road) were converted into raster format for computing the Euclidean distance from each cell to the nearest feature on the basis of spatial analysis in order to connect environmental and artificial factors with landslide susceptibility assessments. A total of 10 landslide-related factors were transferred into the raster format as presented in Table 2. The grid size was 10 ×10 m in this study. In addition, the landslide inventory map was converted into the grid format (10 × 10 m) using different sampling strategies to extract the corresponding landslide factors to connect environmental and landslide information to construct susceptibility models. Regarding rainfall information, it was assumed that landslide occurrences are induced by heavy rainfall and the distribution is constant because the study area is relatively small (~100 km$^2$) with only two rainfall gauge stations located inside the study site. Similar assumptions have been made in other studies [57,77,95–97].

**Table 1.** Landslide inventory information.

|  | Landslide Source | Run-Out |
|---|---|---|
| Number of polygons | 5336 | 1080 |
| Area of maximum polygon (m$^2$) | 250,493 | 385,381 |
| Area of minimum polygon (m$^2$) | 10 | 13 |
| The sum of polygon area (m$^2$) | 10,389,920 | 6,349,667 |
| The average of polygon area (m$^2$) | ~1947 | ~5879 |
| The standard deviation of polygon area (m$^2$) | ~8066 | ~23,355 |

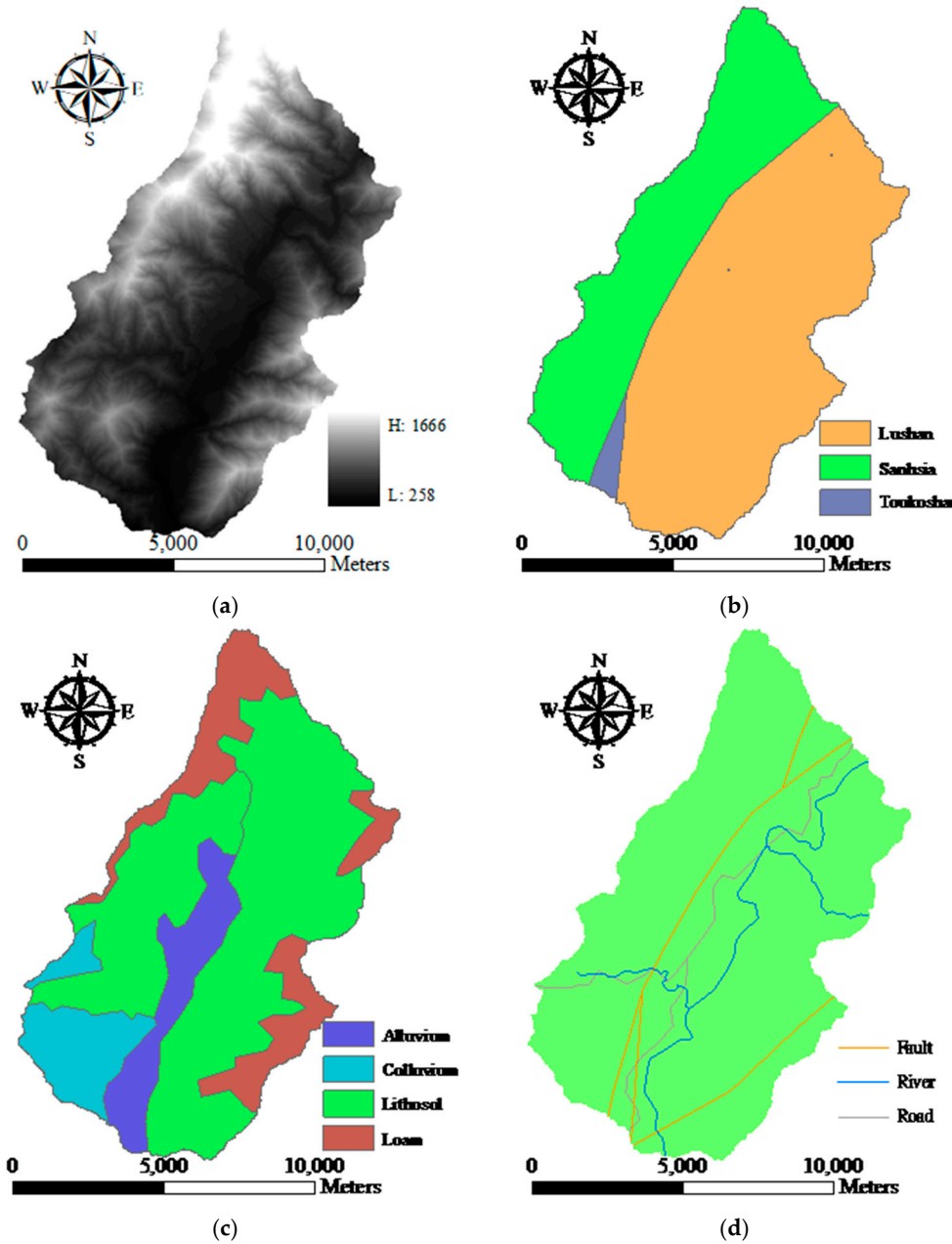

**Figure 2.** *Cont.*

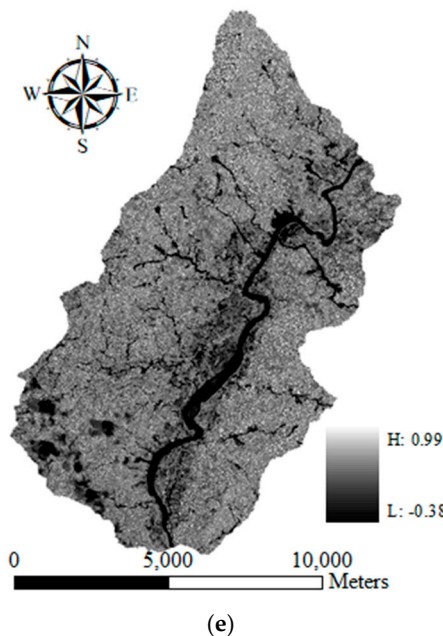

(**e**)

**Figure 2.** Landslide factors used in this study: (**a**) Digital elevation model, (**b**) geological formations, (**c**) soil types, (**d**) line features, (**e**) normalized difference vegetation index image.

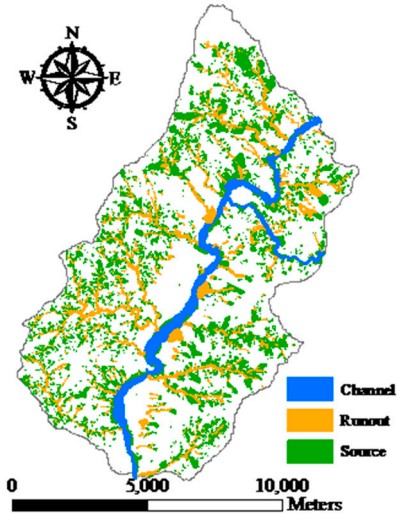

**Figure 3.** Landslide inventory.

**Table 2.** Landslide factors.

| Original Data | Used Factor (Raster Format) |
| --- | --- |
| DEM | Aspect |
|  | Curvature |
|  | Elevation |
|  | Slope |
| Geology map | Geology |
| Soil map | Soil |
| Fault map | Distance to fault |
| River map | Distance to river |
| Road map | Distance to road |
| Satellite imagery | NDVI |

Abbreviations: DEM, digital elevation model; NDVI, normalized difference vegetation index.

*2.2. Developed Procedure*

2.2.1. Sampling Strategies and Analytical Schemes

Implementation of the proposed sampling procedure is illustrated in Figure 4. The landslide inventory is prepared using the centroid method and three statistical operators to generate different sample-sets. Centroid locations are sought polygon by polygon. In addition, for the application of statistical operators, the landslide inventory is overlaid with the slope layer to determine the maximum (Max), median (Med), and minimum (Min) values, polygon by polygon, with the corresponding locations treated as representative samples. This is because the slope distribution is significant in relation to the surface flow velocity, runoff rate, geomorphology, and soil water content [76], and a relationship exists between the landslide source and run-out features.

Data are collected for two combinations of the source and run-out classes. The first strategy is to collect samples for both classes extracted from the same sampling operator. The other strategy is to combine different sampling results (i.e., hybrid strategy). By contrast, non-landslide samples were randomly extracted in an amount set as equal to the sum of the source and run-out instances. Next, the aforementioned combinations were integrated with non-landslide samples to match the landslide factors for preparing training and test datasets.

Three analytical schemes, as listed in Table 3 are used in conjunction with different sampling strategies to construct the landslide susceptibility models. Model-1 is designed to classify the landslide source, run-out, and non-landslide as three separated classes. The run-out class is considered a part of the source class in Model-2 and the non-landslide class in Model-3.

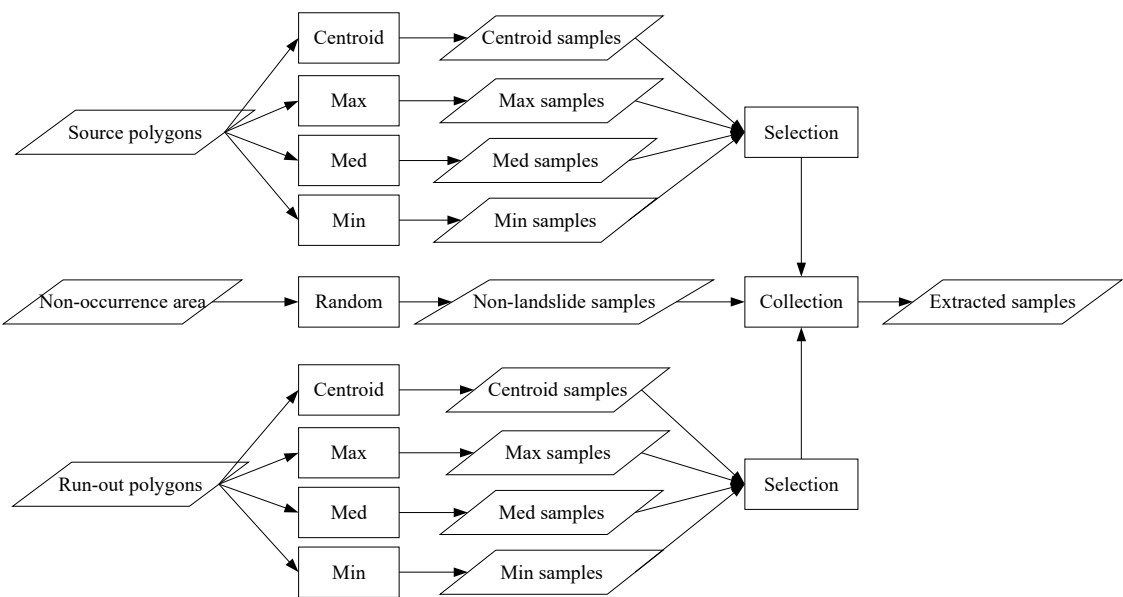

**Figure 4.** Sampling strategy procedure used to extract landslide causative factors to prepare training and check datasets; Max, Med, and Min represent maximum, median, and minimum operators, respectively.

**Table 3.** Three analytical schemes for constructing landslide susceptibility models.

| Scheme | Number of Classes | Notation |
| --- | --- | --- |
| Model-1 | 3 | Run-out is an individual class |
| Model-2 | 2 | Run-out belongs to the landslide source class |
| Model-3 | 2 | Run-out belongs to the non-landslide class |

2.2.2. Random Forests

The RF algorithm [98] is a supervised and nonparametric machine-learning-based method employed in this study to construct landslide susceptibility models. RF is an extension of the decision

tree (DT) algorithm, which is a classical and popular approach in related domains. Both RF and DT classifiers are similar in concept, adopting the information gain (IG) measure or Gini index to evaluate the degree of impurity of factors or variables. A larger IG or Gini value indicates that the corresponding factor should be selected as having a higher priority for construction of a conditional node and this factor should be ignored in the next computation. After several iterations, a tree model consisting of a sequence of rules is extracted, which is used to classify other instances. The difference between RF and DT is that the former randomly separates training data into many subsets using the bootstrap algorithm to build many trees (the so-called "forests") for optimization. This is why more accurate results can be obtained with the RF classifier than the DT algorithm.

In general, the two major geo-spatial data formats are nominal (or discrete) or numeric (or continuous). For discrete data, the information gain is computed using entropy, as described in Equations (1)–(3), where E(A) indicates the entropy of all training data, m is the number of classes, n and N are the subset and total amounts within the decision attribute (i.e., label or class), respectively, $E'(a)$ and v represent the entropy and subset amounts of a specific landslide factor, respectively, $E(a_j)$ is the entropy of the subset in a specific landslide factor computed by Equation (1), and IG(a) indicates the information gain of a specific landslide factor. For continuous data, the Gini index is utilized to calculate information gain, as described in Equations (4) and (5), where C represents a segmented point for a specific landslide factor used to divide numeric data into two parts and $N_1$ and $N_2$ are the numbers of $a \leq C$ and $a > C$, respectively.

$$E(A) = -\sum_{i=1}^{m} \frac{n_i}{N} \log_2 \frac{n_i}{N}. \tag{1}$$

$$E'^{(a)} = \sum_{j=1}^{v} \frac{n_j^a}{N} E(a_j). \tag{2}$$

$$IG(a) = E(A) - E'(a). \tag{3}$$

$$Gini(a \leq C \text{ or } a > C) = 1 - \sum_{i=1}^{m} \frac{n_i}{N}. \tag{4}$$

$$IG(a, C) = \frac{N_1}{N} Gini(a \leq C) + \frac{N_2}{N} Gini(a > C). \tag{5}$$

### 2.2.3. Cost-Sensitive Analysis

Cost-sensitive analysis or learning is a post-classification method that adjusts the decision boundary according to the cost matrix and reclassifies it to increase and balance the accuracies of certain classes when their omission or commission error is unacceptable [90–92]. The size of the cost matrix is the same as the confusion matrix. The terms true positive (TP), false negative (FN), false positive (FP), and true negative (TN) are commonly used when referring to the counts tabulated in a confusion matrix. The diagonal costs in a confusion matrix indicate the cost of correct classification (i.e., TN and TP), and the remainder (i.e., FN and FP) represents the misclassification costs between various classes. These costs are usually set to 0 and 1 for the diagonal and other elements. The decision boundary can be enlarged by increasing the cost of the incorrect parts to include more samples to balance the classification result of a certain class. However, this might have a positive or negative effect on the results of other classes. To address this issue, different costs for the target class are tested for balancing the false alarm and missing rates as well as maintaining the reliability and veracity of the overall results.

For optimal decision-making, the cost matrix should be classified into the class that has the minimum expected cost. The optimal prediction (R) for an instance x in class i can be expressed as in Equation (6), where P(j|x) is the probability for the estimation of a classification of an instance into

class j and C indicates the cost. In the two-class case, the optimal prediction is class 1 if and only if the expected cost of this prediction is less than or equal to the expected cost of predicting class 0, as listed in Equation (7). Given $p = P(1|x)$ and $C_{00} = C_{11} = 0$, Equation (7) is equivalent to Equation (8), and a threshold $p^*$ can be defined, according to Equation (10) based on Equation (9), for the classifier to classify an instance x as positive if $P(1|x)$ is larger than or equal to the threshold [99].

$$R(i|x) = \sum_j P(j|x)C_{ji}. \tag{6}$$

$$P(0|x)C_{10} + P(1|x)C_{11} \leq P(0|x)C_{00} + P(1|x)C_{01}. \tag{7}$$

$$(1-p)C_{10} \leq p\,C_{01}. \tag{8}$$

$$(1-p^*)C_{10} \leq p^*C_{01}. \tag{9}$$

$$p^* = \frac{C_{10}}{C_{10} + C_{01}}. \tag{10}$$

### 2.2.4. Accuracy Assessment and Mapping

After constructing the landslide susceptibility model, the landslide likelihood of an instance can be assessed. The landslide susceptibility map is further generated by inputting test samples for the entire area (called instances) into the model. For the task of verification, the confusion matrix and quantitative indices are derived from comparisons between the output and reference labels (e.g., landslide or occurrence and non-landslide or nonoccurrence) of the check datasets as identified by a threshold. In general, an instance is classified into the landslide category when the landslide likelihood is larger than or equal to 0.5; otherwise, the instance is not given the occurrence label (non-landslide). After adjusting the decision boundary, the threshold is equal to $p^*$.

Overall accuracy, precision/UA (user's accuracy), recall/PA (producer's accuracy), area under the ROCs (receiver operating characteristics) curve, and kappa coefficient derived from a confusion matrix are used as indices for quantitative evaluation and verification of the constructed model in this study. A confusion matrix is used to compare and count the classification results of the check data (output labels) against the ground truth (reference labels). PA and UA reflect the errors class by class that can be calculated according to Equation (11) and Equation (12), respectively, where M represents an element in a confusion matrix, Nc is the number of classes, and i and j are the indices for columns and rows in a confusion matrix individually. Furthermore, the omission and commission errors can be obtained by 1–PA and 1–UA, respectively. Overall accuracy refers to the percentage of correctly classified samples, as shown in Equation (13). The area under the ROCs curve (AUC) is also commonly used [16,38,44,75,77] to assess landslide susceptibility models. The binary classification threshold is changed to reclassify samples and calculate the FP rates (i.e., 1–PA = omission error of the non-landslide class) and TP rates (i.e., PA of the landslide class) to draw the ROCs curve. In general, the AUC ranges from 0.5 to 1. A larger AUC indicates that the result is more reliable. The kappa coefficient is defined in Equation (14) and is used for calculating the proportion of agreement after chance agreement is removed from consideration. A negative kappa value suggests the agreement is worse than would occur from randomly assigning each sample to a class and kappa = 1 indicates perfect agreement. The landslide susceptibility map is generated pixel by pixel according to the landslide likelihoods or intervals (e.g., low, medium, and high susceptibilities) outputted from the developed model when the verified results are acceptable.

$$PA_j = \frac{M_{jj}}{\sum_{i=1}^{N_c} M_{ij}} = \frac{M_{jj}}{M_{+j}}. \tag{11}$$

$$UA_i = \frac{M_{ii}}{\sum_{j=1}^{N_c} M_{ij}} = \frac{M_{ii}}{M_{i+}}. \tag{12}$$

$$OA = \frac{\sum_{i=1}^{N_c} M_{ii}}{\sum_{i=1}^{N_c} \sum_{j=1}^{N_c} M_{ij}} = \frac{M_{diag}}{M_{total}}. \tag{13}$$

$$Kappa = \frac{M_{total}M_{diag} - \sum_{i=1}^{N_c}(M_{+i}M_{i+})}{M_{total}^2 - \sum_{i=1}^{N_c}(M_{+i}M_{i+})}. \tag{14}$$

## 3. Results

A two-stage study was designed to explore the influence of sampling strategies and run-out area on landslide susceptibility modeling. The first stage compared the slope characteristics between the source and run-out samples achieved with different sampling strategies. Furthermore, the constraint on area size for inventory polygons was considered. In the second stage, the three analytical schemes listed in Table 3 were used in conjunction with different sampling strategies to construct the landslide susceptibility models. This study used the RF algorithm as well as the accuracy assessment in the WEKA program (http://www.cs.waikato.ac.nz/mL/weka/), which is a free and open-source platform. The number of trees (Ntree) is the major parameter in WEKA for RF computation. Du et al. [82] indicated that results using a number of trees from 10 to 200 have no influence of prediction capability. Computation loading is also increased by the extension of Ntree. Thus, this study adopted 100 trees, as suggested by WEKA, to perform RF-based landslide susceptibility assessments.

### 3.1. Topographic Characteristics

To preliminarily examine the effect of the sampling strategy, the slope distributions were compared in light of the difference between the landslide source and run-out classes as well as the constraint on area size of the inventory polygons. The samples were extracted according to different area sizes: equal to or more than 100, 250, 500, 750, and 1000 $m^2$. The numbers of extracted samples are listed in Table 4. Notably, the number of samples in Table 4 without an area constraint is less than the number of polygons in Table 1. The data preprocessing step transforms the inventory polygons into pixel-based samples by using different sampling strategies to extract the corresponding landslide factors. Some source and run-out samples based on maximum, median, and minimum sampling operators may be located beyond the study area after transformation from vector to raster format and thus yield no data. Therefore, this study ignores those unavailable samples. The results obtained by subtracting the average slope of the run-out from the source class are shown in Figure 5a. It is obvious that a larger area size enables the source and run-out classes to be distinguished better. Moreover, the centroid and median sampling operators vary more with the inventory polygon size. To maintain sufficient distinguishability and a sufficient number of samples, the samples were extracted from inventory polygons with an area equal to or larger than 1000 $m^2$ for further analysis. The area size, which was smaller than the previous threshold, was also extracted for comparison. Figure 5b shows that it is more difficult to distinguish between the source and run-out classes when using the small area threshold (area <1000 $m^2$). Furthermore, the average slope of the source areas was larger than the run-out class with both constraints. Figure 5c presents the standard deviations for Figure 5b. Both Figure 5b,c suggest that the slope distributions are lower and there are more variations for the run-out class, which is reasonable, because these samples contain landslide trails and depositions, resulting in mixed results.

**Table 4.** Numbers of extracted source and run-out samples with different area constraints.

| Constraint on Area Size | Source | Run-Out |
|:---:|:---:|:---:|
| ≥1000 $m^2$ | 1345 | 445 |
| ≥750 $m^2$ | 1638 | 536 |
| ≥500 $m^2$ | 2078 | 638 |
| ≥250 $m^2$ | 2943 | 821 |
| ≥100 $m^2$ | 4049 | 981 |
| No constraint | 4656 | 1020 |

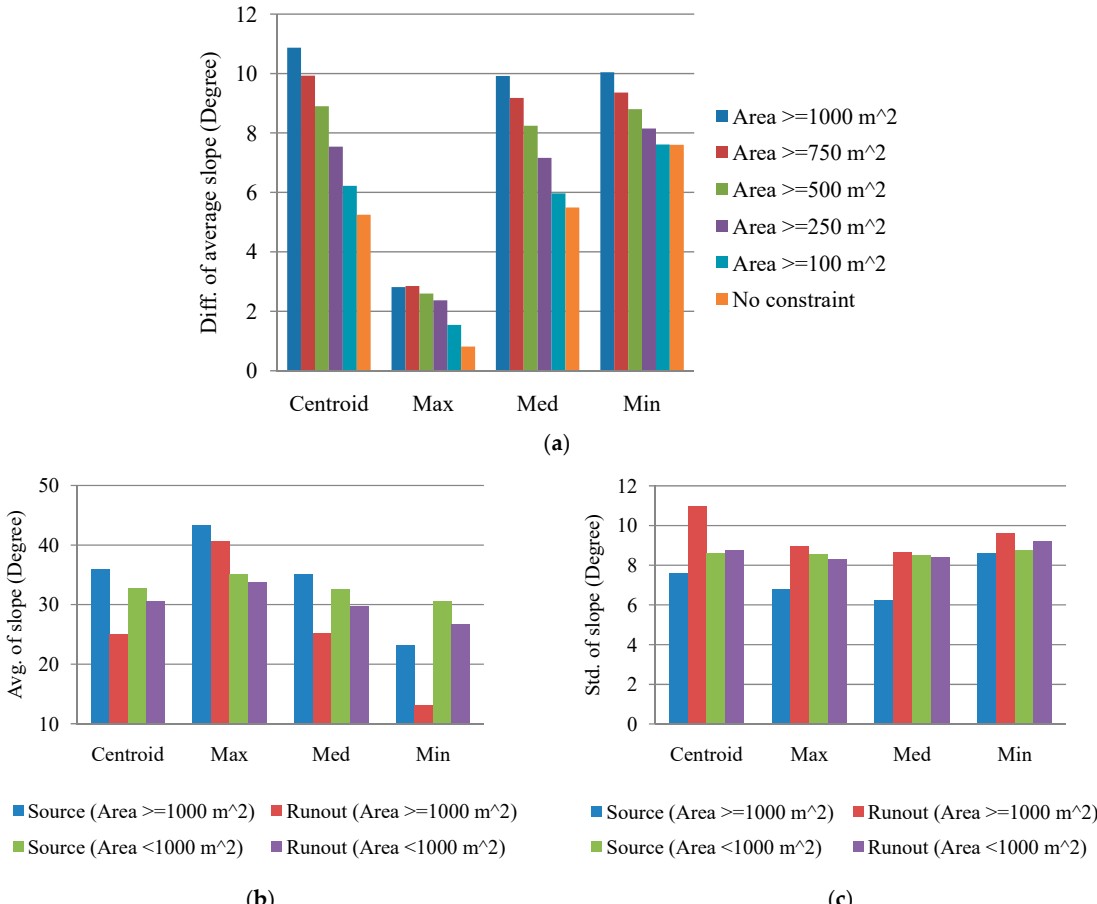

**Figure 5.** Slope distributions between the source and run-out areas using different sampling operators: (**a**) Difference in the average slope according to subtracting the run-out area from the source region in consideration of area constraints, (**b**) average slope of the area size of ≥ and <1000 m$^2$, (**c**) standard deviation of (b).

*3.2. Modeling Performance*

To further explore the feasibility of modeling the run-out area with different sampling strategies and area constraints, landslide susceptibility models were constructed using Model-1 to identify the source, run-out, and non-landslide signatures. In addition, non-landslide samples were randomly extracted with the amount set to be the same as the sum of the source and run-out instances. Figure 6 shows the results of the evaluation of Model-1 using different sampling operators with or without the area constraint. The results demonstrate that consideration of the constraint on the area size is important not only because it helps distinguish the topographic characteristics between the source and run-out area but, in most situations, it also increases the classification accuracy of the RF algorithm, especially the run-out's UA and PA. The results also indicate that the modeling of the non-landslide and source classes is reliable and that the maximum operator in particular provides higher accuracy. However, run-out omission and commission errors are relatively high. The hybrid sampling strategy, in which the source and run-out samples that meet the area constraint of ≥1000 m$^2$ are extracted by various sampling operators, is further considered. Figure 7 illustrates the Max–Min sampling strategy, in which the maximum and minimum operators are applied to extract the source and run-out samples, respectively, outperforms other combinations when taking all classes into account (the confusion matrix result is shown in Table 5). These findings correlated well with those in Figure 6, which indicates that the maximum and minimum operators perform better for the source and run-out classes individually. Compared with logistic regression [25–27], a commonly used algorithm in related domains, Table 5 also demonstrates the efficiency of RF.

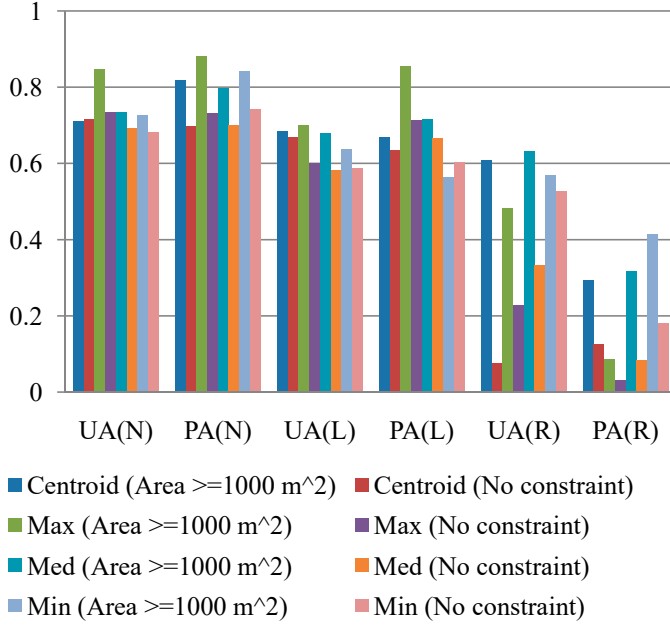

**Figure 6.** Quantitative evaluations of Model-1 using different sampling operators to compare the results of applying the constraint on the area size or not. UA—user's accuracy, PA—producer's accuracy, N—non-landslide class, L—landslide source class, R—run-out class.

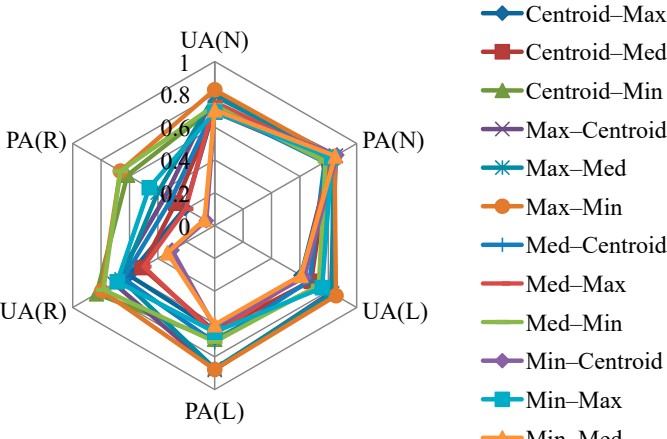

**Figure 7.** Quantitative evaluations of Model-1 using the hybrid sampling strategies and the area constraint (≥1000 m$^2$). For example, Centroid–Max indicates that the source and run-out samples are extracted by the centroid and maximum operators, respectively.

The run-out area is also treated as part of the landslide source and non-landslide classes (Model-2 and Model-3) using the Max–Min sampling strategy. The classification evaluations are illustrated in Figure 8. They indicate that the run-out area increases the source's omission error but improves the non-landslide's UA and PA (Model-3). The results indicate that ignoring the difference between the source and run-out signatures (i.e., Model-2) when producing a landslide inventory will lead to a degradation in the accuracy of the susceptibility modeling. It is suggested that the run-out area should be considered as part of the non-landslide class if the run-out's UA and PA from Model-1 are not acceptable when treated as an independent class. As evident in Figure 8, the run-out's omission error is still high. To address this issue, cost-sensitive analysis is performed to adjust the decision boundary during classification. According to Table 5, high omission error occurs because of misclassification between the run-out and non-landslide areas. The cost of misclassification is thus set to five- and 10-fold (referred to as C=5 and C=10) in the cost matrix, and they are reclassified. Figure 9 compares

the classification results of the original condition (C=1) with these cost settings. When C=5, the best trade-off between UA and PA is obtained for the run-out class and the overall results are maintained.

**Table 5.** Confusion matrix result of Model-1 using the Max–Min sampling strategy without the cost setting.

|  |  | Ground Truth | | | |
| --- | --- | --- | --- | --- | --- |
|  |  | Source | Run-Out | Non-Landslide | UA |
| | | (a) Random Forests | | | |
| Prediction results | Source | 408 | 1 | 69 | 0.854 |
| | Run-out | 1 | 101 | 25 | 0.795 |
| | Non-landslide | 56 | 50 | 506 | 0.827 |
| | PA | 0.877 | 0.664 | 0.843 | |
| Overall Accuracy = 83.4% | | | | Kappa = 0.7182 | |
| | | (b) Logistic Regression | | | |
| Prediction results | Source | 361 | 1 | 81 | 0.815 |
| | Run-out | 1 | 97 | 1 | 0.688 |
| | Non-landslide | 103 | 54 | 476 | 0.752 |
| | PA | 0.776 | 0.638 | 0.793 | |
| Overall Accuracy = 76.7% | | | | Kappa = 0.6059 | |

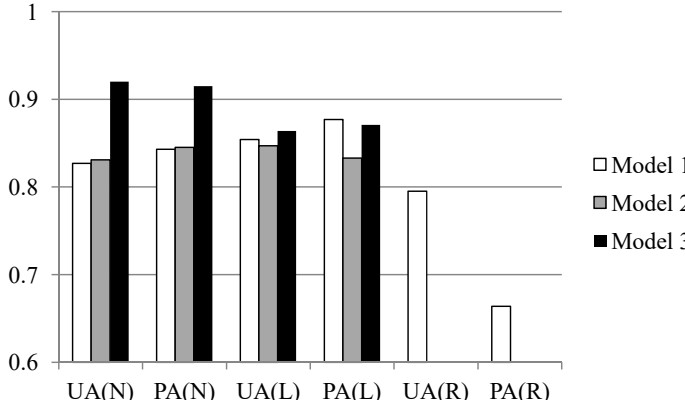

**Figure 8.** Quantitative evaluations of treating the run-out area as the landslide source (Model-2) and non-landslide (Model-3) classes compared with Model-1 using the Max–Min sampling strategy.

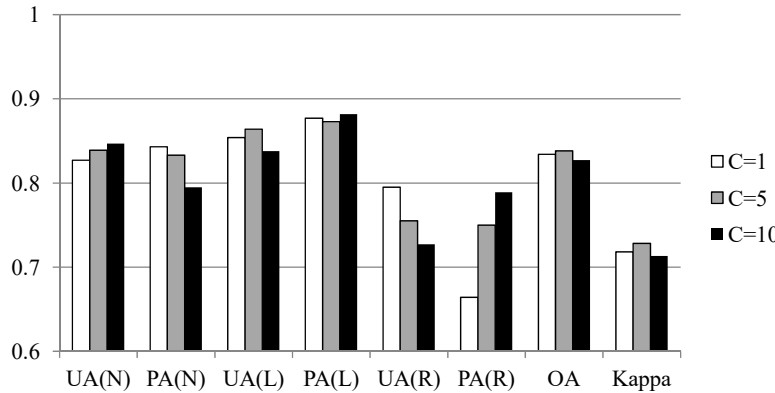

**Figure 9.** Quantitative evaluations of Model-1 using different cost settings to balance the run-out's omission and commission errors. C—cost weight in a cost matrix, OA—overall accuracy, Kappa—kappa coefficient.

The 2010, 2011, and 2015 landslide records were used for evaluation and comparison against the ground truth to assess the prediction accuracy of the developed models. These records included the expansion of earlier landslides triggered by Typhoon Morakot. The feasibility of tracking a similar landslide type (deep-seated) was tested by applying the same sampling strategy (Max–Min) and constraint (area≥1000 m$^2$) to the 2010, 2011, and 2015 landslide records. Table 6 shows the number of extracted source and run-out samples. Moreover, non-landslide samples were extracted randomly with amounts set to be the same as the sum of source and run-out instances.

**Table 6.** Numbers of extracted source and run-out samples of later records.

| Year | Source | Run-Out |
|------|--------|---------|
| 2010 | 361 | 550 |
| 2011 | 296 | 520 |
| 2015 | 205 | 85 |

The performance of Model-1 is presented in Table 7. It is clear that the 2010 and 2011 results are similar but the accuracies are worse than those in Table 5, especially for the run-out's PA. Although the 2015 results are more accurate, the run-out omission error is still high. These results suggest that there is uncertainty in different events and sampling ratios (Tables 4 and 6). However, these results are based on a fixed classification threshold. Different classification thresholds were considered. The AUC evaluator was adopted to explore the flexibility of the developed models. The AUC results in Table 7 indicate acceptable accuracy. These prediction capabilities may serve as the basis for a study of run-out detection. They also indicate the prediction limitations of Model-1 because the separability between the landslide source, run-out, and non-landslide is not absolutely clear. Addressing this issue will require further improvement of the developed models, such as by the integration of physical or topographic parameters and field data.

**Table 7.** Prediction of Model-1 using the later-record samples. AUC—area under the receiver operating characteristics curve.

| Year | Non-Landslide | | | Source | | | Run-Out | | |
|------|------|------|------|------|------|------|------|------|------|
| | UA | PA | AUC | UA | PA | AUC | UA | PA | AUC |
| 2010 | 0.654 | 0.698 | 0.721 | 0.613 | 0.839 | 0.932 | 0.697 | 0.451 | 0.814 |
| 2011 | 0.652 | 0.688 | 0.717 | 0.567 | 0.855 | 0.92 | 0.706 | 0.442 | 0.795 |
| 2015 | 0.819 | 0.734 | 0.857 | 0.732 | 0.878 | 0.909 | 0.649 | 0.565 | 0.843 |

Model-2 and Model-3 were also used to perform landslide predictions. The UAs and PAs in Table 8 indicate that the run-out characteristics are consistent with the non-landslide signature because Model-3 outperformed Model-2. In addition, the results for 2015 without cost setting are greater than 0.8 in most cases. Cost-sensitive analysis was also applied for improving 2010 and 2011 results. Table 8 illustrates the reduction in source commission errors by cost setting. Therefore, it is critical to separate the run-out area from the landslide-affected area detected through automatic and semiautomatic approaches.

*3.3. Landslide Susceptibility Mapping*

To visualize the modeling results, landslide susceptibility maps were generated, as shown in Figure 10a–c. As can be seen in the figures, the spatial patterns of Model-1 (with C=5) and Model-3 are similar but are both quite different from those of Model-2. Figure 10d further illustrates the difference between Model-1 and Model-3 determined by subtracting the data in Figure 10c from that in Figure 10a. As displayed in Figure 10d, although the patterns of Model-1 and Model-3 are similar, significant differences (–0.49 to 0.43) are observed in the calculated landslide susceptibility in individual pixels. Figure 10e,f further show the black area in Figure 10b,c, revealing that Model-2 produces overestimated

results with salt-and-pepper noise. Although the landslide inventory is of an object-based format, it should be noted that after extracting samples polygon by polygon, the developed model was processed using record-based (or pixel) computation. This may have also contributed to the salt-and-pepper effect.

**Table 8.** Prediction of Model-2 and -3 using later-record samples.

| Year | | AUC | Non-Landslide | | Source | |
|---|---|---|---|---|---|---|
| | | | UA | PA | UA | PA |
| (a) Without cost setting | | | | | | |
| 2010 | Model-2 | 0.869 | 0.917 | 0.729 | 0.549 | 0.834 |
| | Model-3 | 0.912 | 0.935 | 0.822 | 0.656 | 0.856 |
| 2011 | Model-2 | 0.871 | 0.929 | 0.708 | 0.514 | 0.851 |
| | Model-3 | 0.897 | 0.936 | 0.801 | 0.608 | 0.848 |
| 2015 | Model-2 | 0.881 | 0.9 | 0.748 | 0.713 | 0.883 |
| | Model-3 | 0.908 | 0.912 | 0.824 | 0.781 | 0.888 |
| (b) With cost setting | | | | | | |
| 2010 | Model-2 | 0.869 | 0.898 | 0.84 | **0.652** | 0.759 |
| | Model-3 | 0.905 | 0.902 | 0.885 | **0.722** | 0.756 |
| 2011 | Model-2 | 0.868 | 0.904 | 0.82 | **0.605** | 0.76 |
| | Model-3 | 0.891 | 0.907 | 0.865 | **0.671** | 0.757 |

## 4. Discussion

The effectiveness of automatic statistical sampling methods and cost-sensitive analysis as well as the influence of the run-out area were demonstrated for the study site. These achievements also reveal the significance of landslide inventory in which the considered classes and completeness affect the quality and development of consequent tasks within the landslide risk assessment and management framework. More precisely, the run-out area and topological relationship may be considered independent categories in the landslide records when producing a landslide inventory. The primary limitation of this study is that the constructed models are only suited to the regional scale and may be difficult to apply to other sites. In other words, the outcomes are empirical rather than knowledge based. The landslide susceptibility determined in this study is the spatial likelihood of landslide occurrence instead of the probability. The reason for this is that general data-driven (include data mining) algorithms directly explore the relationship between the collected landslide factors and inventory, and the generalization of these outcomes is a general rule that should be further evaluated. However, the study cases demonstrated the feasibility of the proposed procedure, suggesting that the developed procedure can be used for other sites. The generated landslide susceptibility maps can also be used for land planning, disaster insurance, and rescue purposes.

Previous works have focused on the modeling of landslide and non-landslide labels with ratios of 1:1 to 1:10 [100]. However, the effect of number of run-out samples in the landslide susceptibility modeling process has not been discussed. Regarding the class ratio of classes, this study utilized all the landslide source and run-out samples extracted from the landslide inventory. The amount of non-landslide samples was set to be equal to the sum of the landslide source and run-out classes. Perhaps the impact of the number of samples and labels can be examined further in future.

Landslide inventory is a critical factor in the landslide risk assessment and management framework, and its labels (i.e., landslide source, run-out, and non-landslide), completeness, and topological relationships will increase the value to subsequent tasks. It may be of interest for future research to apply object-oriented classification [62,63] to preserve the completeness and topological relationships of landslide-affected regions and to develop a semi-automatic or automatic approach to further extract run-out areas from these regions and generate the complete landslide inventory. After that, the run-out patterns derived from the landslide susceptibility models may be adequate for further evaluating

related parameters such as run-out distance and damage corridor width, as described in Dai et al. [10]. Finally, future research may also examine the effect of the quality, number of classes (labels), and sample ratios of landslide susceptibility models on the consequent tasks within the landslide risk assessment and management framework.

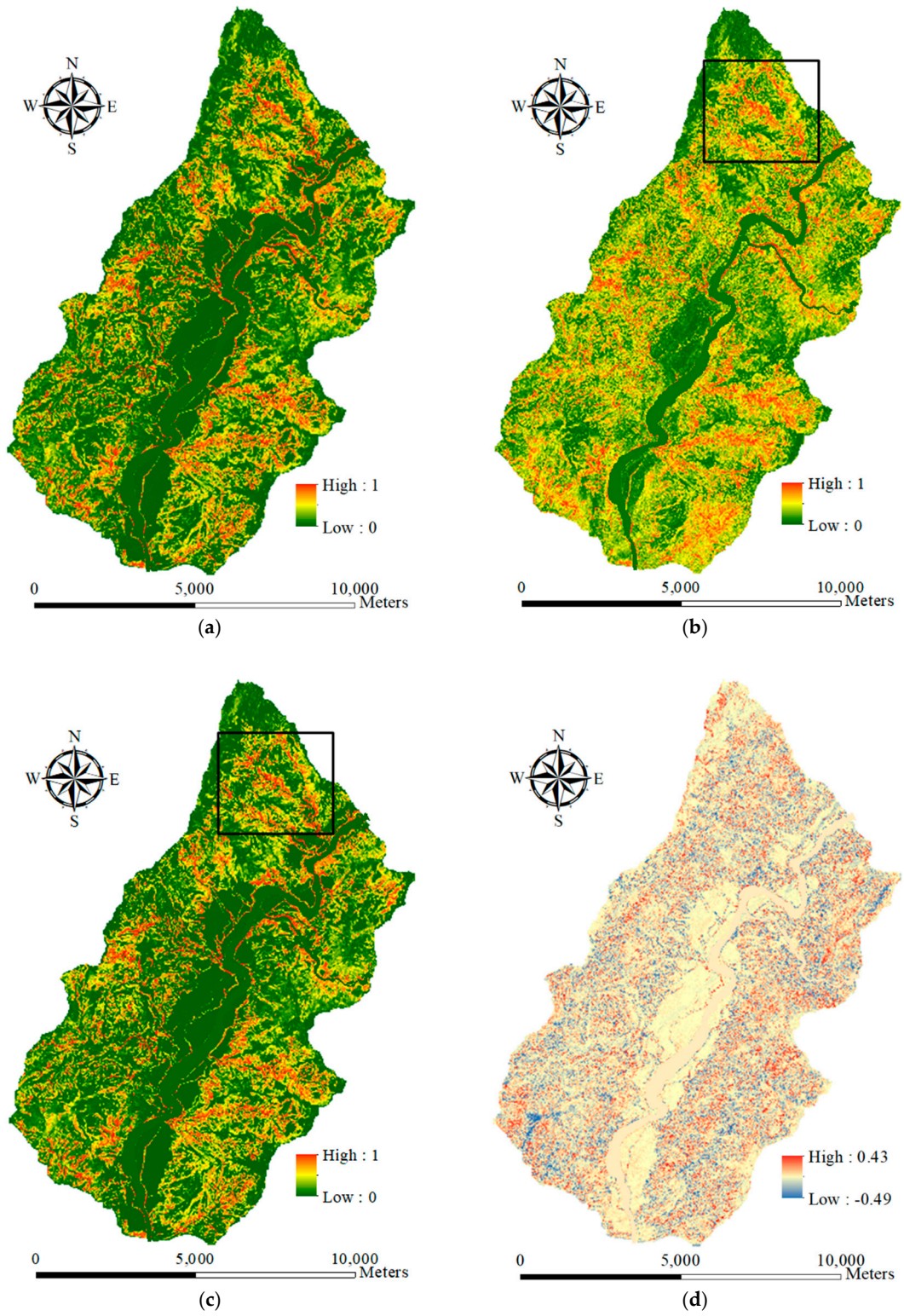

**Figure 10.** *Cont.*

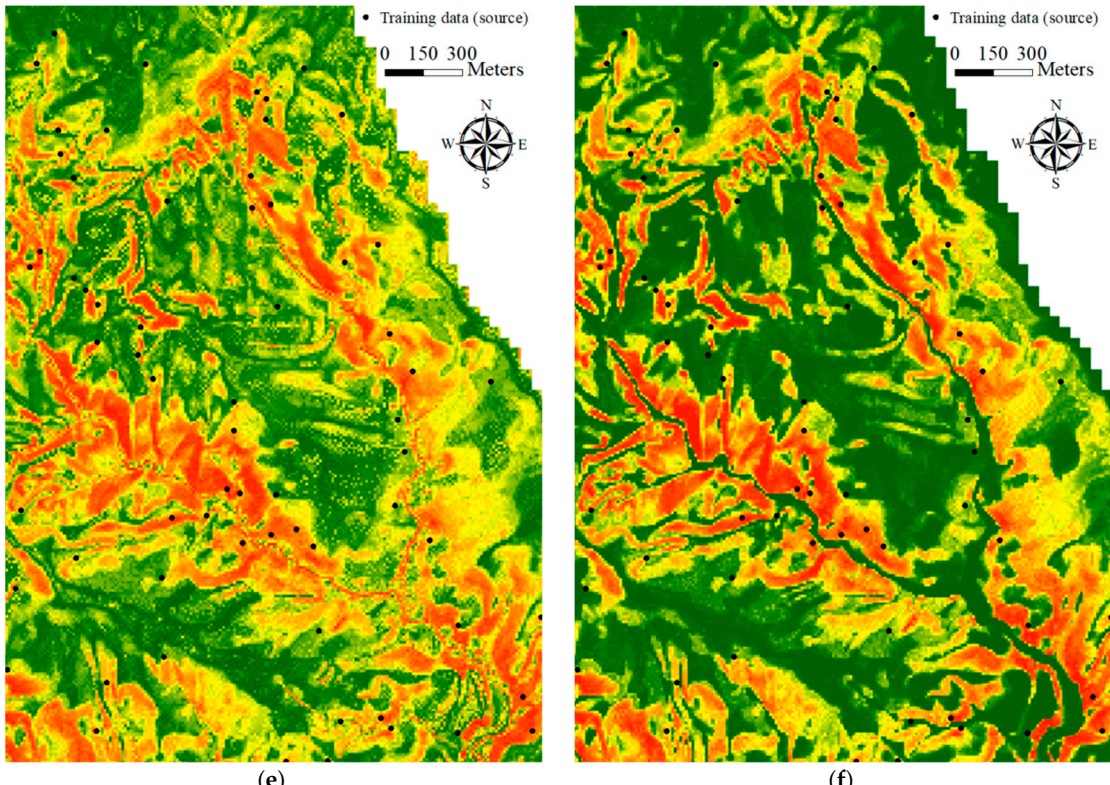

| (e) | (f) |

**Figure 10.** Generated landslide susceptibility maps: (**a**) Model-1 with cost setting (C=5), (**b**) Model-2, (**c**) Model-3, (**d**) difference of subtracting (c) from (a), (**e**) zoomed-in Model-2 result, (**f**) zoomed-in Model-3 result.

## 5. Conclusions

In this study, a procedure was developed for integrating geo-spatial data and machine learning for event-based landslide susceptibility modeling and assessments at a watershed scale. In addition, this study also demonstrated the effect of the sampling strategy and run-out area on the process of landslide susceptibility modeling based on the GIS-based landslide inventory. Experimental results indicated that the topographic characteristics of geo-spatial data, the sizes of the landslide inventory polygons, and the sampling strategies can be used to distinguish between landslide source and run-out areas. Identifying the landslide run-out area as a single class during the landslide susceptibility modeling process (Model-1) using a hybrid sampling strategy (Max–Min) and the area constraint ($\geq 1000$ m$^2$) with RF can outperform logistic regression and achieve better results: Higher than 80%, 0.7, 0.79, and 0.66 for the overall accuracy, kappa, UA, and PA measures, respectively. In addition, by treating the run-out area as a landslide or non-landslide class, it was determined that the run-out area should be considered part of the non-landslide class if treating the run-out as an independent class (Model-1) does not yield acceptable results. Cost-sensitive analysis was also used to adjust the decision boundary to improve Model-1 performance, achieving a 9% increase in the run-out's PA. The results of verifying later landslide events also indicate that using cost-sensitive analysis can lead to an improvement of range from 5% to 10% for the landslide source's UA performance. According to these analyses, it is suggested that run-out should be included as an individual class in a landslide inventory for the construction of more reliable and flexible susceptibility models.

**Author Contributions:** Jhe-Syuan Lai, Shou-Hao Chiang, and Fuan Tsai conceived and designed this study; Jhe-Syuan Lai performed the experiments and analyzed the results; Jhe-Syuan Lai, Shou-Hao Chiang, and Fuan Tsai wrote the paper.

**Funding:** This study was supported, in part, by the Ministry of Science and Technology of Taiwan under the project 108-2119-M-035-003.

**Conflicts of Interest:** The authors declare that they have no conflicts of interest.

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
