# Peer review of "Exploring Influence of Sampling Strategies on Event-Based Landslide Susceptibility Modeling"

_ijgi, doi:10.3390/ijgi8090397_

Round 1

Reviewer 1 Report

This study explored two modeling issues that may cause uncertainty in landslide susceptibility assessments: (1) how to extract attributes within a landslide polygon; (2) the mixing problem of landslide inventory. To this end, the authors designed a two-stage experiment to explore the influence of different statistical sampling strategies (Centroid, Max, Med and Min) and run-out samples on landslide susceptibility modeling based on random forests. In addition, the cost sensitive analysis was applied to improve the models with extremely high false alarm error or missing error. The experimental results were compared with a logistic regression and evaluated by means of overall accuracy, precision, recall, AUC value and Kappa coefficient.

The manuscript appears to make some contributions to landslide susceptibility assessments. However, there are still some problems needed to be improved.

1. In the introduction, the authors seem to have done a lot of literature research on landslide susceptibility modeling. But the introduction should outline the problem the study is trying to address and highlight the importance and innovation of the study. The method and the main conclusions should also be briefly mentioned. Authors might consider combining the introduction and the section 2.

2. Pg. 3 line 117: Random forest method is a commonly used machine learning method on landslide susceptibility modeling. This may be too subjective.

3. Pg. 6 line 182: 10 landslide factors used in the study could be displayed with figures.

4. Pg. 6 line 195: In this study, non-landslide samples were just randomly selected from the nonoccurrence area. Some false non-landslide samples with high landslide susceptibility maybe selected.

5. Pg. 6 line 197: “training and test datasets” may be more common used.

6. Pg. 10 line 315: Why the numbers of the source and run-out samples with no constraint in Table 4 are less than that in Table 1?

7. Pg. 12 line 356: There seems to have a display problem with Figure 6 using Adobe Reader software. It is hard to read with only the grayscale difference. There is no difference between the object with no constraint. Perhaps color will be helpful to make it more readable.

8. Pg. 11 line 333: Please supplement that the hybrid sampling strategy was applied with the area constraint (≥ 1000 m2) in the text.

9. Pg. 11 line 339: Please insert the reference for logistic regression.

10. Pg. 15 line 423: Why there is no results for 2015 landslide in Table 8b?

Reviewer 2 Report

Dear Authors, I've carefully read your manuscript "Exploring Influence of Sampling Strategies on Event-based Landslide Susceptibility Modeling" and I found it interesting, well written and well centered n the topics of the journal (and special issue).

Before publication, I recommend some revisions to improve further the quality of your work.

GENERAL COMMENTS

- I suggest a small adjustment to the manuscript structure. The texts starts with a focus ont he study area: this is typical of technical reports, not of scientific papers. I suggest to start the manuscript with the second part of Introduction (lines 48-71), merged with section 2 into a single section (1. Introduction). The lines from 31 to 46 could be moved to the test site description. This is actually just a cut & paste, no big deals, but it will give the right perspective to your research.

- I think some clarifications are needed about landslide description. Sometimes you declare to deal with deep-seated landslides, sometimes you openly make references to debris flows (e.g. L36), sometimes you describe issues that are commonly related to rapid slides or rapid flows.

- Thee is only a point of the procedure that I didn't find appropriate, and it is at lines 166-167. I don't understand how a parameter "distance form linear features" could improve the susceptibility assessment. Linear features are very different, with very different impacts on landslide activity: a fault, a river, a road, they all may affect the hillslope stability, but they do it in different ways, I don't get the point of putting them together. Indeed, in many works they are treated as separate factors. Either you provide a good justification, or you show a quantitative evaluation of the importance of this "condensed" parameter (maybe comparing statistics that can be used to rank the importance of the parameters, like e.g. the out of bag error).

SPECIFIC COMMENTS

L14: it is not clear to me what you mean with "topographic location". Can you be more specific?

L19: the sentence reads like you are actually trying to obtain a model that prodces many false alarms or missed alarms. Also at line 148. I guess it is the opposite...

Abstract: it seems to me that the end of the abstract accoints only for the results of the second issue (run-out). What about the first one (sampling strategy)? 

L33: please convert in USD, for a more straightforward understanding from international readers

L108: scarp?

L117-118: Actually, I don't think this is true nowadays. I suggest rephrasing. MAybe you could say that RF is a technique that only recently has been applied to landslide sucepètibility but has drawn grat attention and is now quite established. I suggest making reference to some early and recent work about RF in landslide susceptibility, such as Brenning 2005; Youssef et al 2016; Lagomarsino et al 2017; Segoni et al., 2018)

Brenning, A. (2005). Spatial prediction models for landslide hazards: review, comparison and evaluation. Natural Hazards and Earth System Sciences, 5, 853–862.

Lagomarsino, D., Tofani, V., Segoni, S., Catani, F., & Casagli, N. (2017). A tool for classification and regression using random forest methodology: applications to landslide susceptibility mapping and soil thickness modeling. Environmental Modeling & Assessment, 22(3), 201-214.

Segoni, S., Tofani, V., Rosi, A., Catani, F., & Casagli, N. (2018). Combination of rainfall thresholds and susceptibility maps for dynamic landslide hazard assessment at regional scale. Frontiers in Earth Science, 6, 85.

Youssef, A. M., Pourghasemi, H. R., Pourtaghi, Z. S., & Al-Katheeri, M. M. (2016). Landslide susceptibility mapping using random forest, boosted regression tree, classification and regression tree, and general linear models and comparison of their performance at Wadi Tayyah Basin, Asir Region, Saudi Arabia. Landslides13(5), 839-856.

L141: please revise if the words ""and run out" at the end of the sentence are necessary.

L196: I think you should state how many of them you selected.

L284: I suggest to nae this section "Results" and section 4.4 could become section 5.

L293: compared to all the studies I am aware of, 10 seems not enough.

I don't see figure 6. Maybe it is an issue of my computer? 

L435-439. I like the outcomes of your study, I would give them more emphasis. Just say that you demostrated something in a case study and that this coiul be of help for future application. Of course, to generalize these outcomes as a general rule, further test are needed.

Round 2

Reviewer 1 Report

The problems raised earlier have been modified accordingly. The manuscript can be published in the present form.